# Percentiles and Reference Values for the Accelerometric Assessment of Static Balance in Women Aged 50–80 Years

**DOI:** 10.3390/s20030940

**Published:** 2020-02-10

**Authors:** Raquel Leirós-Rodríguez, Vicente Romo-Pérez, Jose Luis García-Soidán, Jesús García-Liñeira

**Affiliations:** 1Faculty of Physical Therapy, University of Vigo, Campus a Xunqueira, s/n, 36005 Pontevedra, Spain; 2Faculty of Education and Sport Sciences, University of Vigo, Campus a Xunqueira, s/n, 36005 Pontevedra, Spain; vicente@uvigo.es (V.R.-P.); jlsoidan@uvigo.es (J.L.G.-S.); jeslinheira@alumnos.uvigo.es (J.G.-L.)

**Keywords:** balance, biomechanical phenomena, falls, functional performance, geriatric assessment, physical therapy specialty

## Abstract

The identification of factors that alter postural stability is fundamental in the design of interventions to maintain independence and mobility. This is especially important for women because of their longer life expectancy and higher incidence of falls compared to men. The objective of this study was to construct the percentile box charts and determine the values of reference for the accelerometric assessment of the static balance in women. For this, an observational and cross-sectional study with a sample composed of 496 women (68.8 ± 10.4 years old) was conducted. The measurement of accelerations used a triaxial accelerometer during three tests: two tests on the ground in monopodal support and a test on a mat with monopodal support for 30 s each. In all of the variables, an increase in the magnitude of the accelerations was detected as the age advanced. The box charts of the percentiles of the tests show the amplitude of the interquartile ranges, which increased as the age advanced. The values obtained can be used to assess changes in static balance due to aging, trauma and orthopaedic and neurodegenerative alterations that may alter postural stability and increase the risk of falling.

## 1. Introduction

Each year, one in three adults over 65 years of age and one in two over 80 years of age will suffer a fall [1,2]. Falls cause moderate to severe injuries in 30% of cases; in the elderly, this results in fractures, functional deterioration, a reduction of physical activity levels, premature entry into residential care institutions, fear of falling and even death [3,4,5]. The fear of falling (which is sometimes present without having experienced previous falls) is a serious consequence, since it leads to a cyclical pattern of reduced mobility, social isolation and diminished quality of life [4,6]. The identification of factors that alter postural stability is fundamental in the design of interventions to maintain independence and mobility. This is especially important for women because of their longer life expectancy and higher incidence of falls compared to men [7].

Postural control is related to the centre of gravity (CG), which, according to Mapelli et al. [8], is the result of the multi-segmental conception of equilibrium. That is, it is the conception of the body as a system of rigid bodies whose CG is the average of all of the centres of mass of these segments, a definition that follows the line proposed by Hogdes et al. [9]. Consequently, CG control is part of the requirements for the maintenance of balance during activities of daily life, which include fundamental daily activities such as walking, going up and down stairs, stooping or performing sit-down and standing transfers, and vice versa [10]. Traditionally, balance has been assessed qualitatively (and, to a greater or lesser degree, subjectively) in the clinical setting and quantitatively in laboratory environments through the use of force platforms and dynamic computerised posturography. The force platforms compare the displacement of the centre of pressure (CP) between the feet, which is an independent parameter to the CG; that is, it is a parameter which is strongly conditioned by the intrinsic activity of the ankle and an object of study with the inverted equilibrium pendulum theory [11]. However, this theory, which is valid for the study of some movement strategies in which the postural control system for the maintenance of balance is important, is unsuitable for a complete evaluation of the functioning of the postural control system and of all the strategies from which it is used to maintain balance [12,13]. However, kinematic instruments such as accelerometers allow the objective study of equilibrium, by analysing the CG without the need for a large economic investment in the devices or complex and extensive data processing and analysis processes [14,15]. On the other hand, a model of dynamic computerised posturography widely undertaken in research is EquiTest^®^ (NeuroCom International Clackamas, United States), which does not provide information on the difficulty that an individual has maintaining or controlling posture, but only quantifies the degree of functional limitation for standing, allowing to predict the risk of falling and to evaluate rehabilitation programs in relation to three subsystems (vestibular, somato-sensory and visual) [16,17].

In recent years, several review articles have summarised progress in the use of this tool, and all of them have highlighted the wide variety of analyses and variables used to quantify postural stability [14,18]. At the same time, they also indicate the need to identify a set of accelerometric variables in order to define a solid, objective and reliable model for the clinical assessment of equilibrium [19,20,21]. In this line, the studies of Leirós-Rodríguez et al. [22,23] have advanced, in which the reliability of the records from the fourth lumbar vertebra was confirmed, the suitability of the realisation of the static equilibrium tests for 30 s was assessed and the creation of a tool composed of three static equilibrium tests was reported [24]: two tests on the ground in monopodal support (one with open eyes and one with closed eyes) and a test on a mat with monopodal support with open eyes. From the accelerometric records during the same period, eleven variables should be drawn, all of which refer to the maximum and/or averaged values obtained in the vector module, in the sagittal plane (for the test of monopodal equilibrium on mat) and the frontal plane (for the tests of monopodal balance on the ground).

An integral geriatric assessment, where the balance is an important parameter to be evaluated, would allow us to identify people who are at risk of falls and other adverse pathologies related to balance. Early knowledge of these risk groups, which are far from the average reference values, would allow us to optimise the action measures (thus improving decision-making), control and evaluate the effects of the intervention programs and, at a general level, help to plan public health policies that allow the allocation of resources for the introduction of accelerometers in health consultations, where it would be possible to determine which population can benefit from early preventive interventions related to balance.

Therefore, the objective of this study was to provide normative data for postural stability in elderly women using percentile box charts and determine the values of reference for the accelerometric assessment of the static equilibrium in women according to the tool designed by Leirós-Rodríguez et al. [24]. The percentiles obtained are of interest for the evaluation of static equilibrium in older women. Likewise, the percentiles allow the possibility of making projections of functionality and balance in the medium and long term, which are useful for predicting the risk of falling at certain ages.

## 2. Materials and Methods

### 2.1. Sample

An observational and cross-sectional study was carried out in a random sample of 496 adult women from the city of Ourense (Spain). All of them were recruited from municipal sports centres. The following inclusion criteria were used: (a) engaged in physical activity between 1 and 2 days/week; (b) walked between 30 and 90 min 4 days a week; (c) be over 50 years; and (d) have a good level on independence and gait stability (could complete the Timed Up and Go Test in 10 s or less) [23].

The exclusion criteria were: (a) the inability to walk independently; (b) use of external orthopaedic elements to maintain bipodal static balance with eyes open for 60 s; (c) the presence of any contraindication or illness that prevented evaluation using any of the tests/procedures employed in this study; and (d) a history of falling in the past year. This procedure is detailed in Figure 1.

### 2.2. Procedure

The measurement of accelerations used a triaxial accelerometer (ActiGraph LLC, USA). This accelerometer stored a time series of acceleration data in a non-volatile flash memory. The small dimensions of the module (4.6 cm × 3.3 cm × 1.5 cm), combined with its low weight (19 g), accuracy (3 mg/LSB) and range (±6 G), make this device a good choice to evaluate changes in body position in an outpatient environment. 

Accelerometers provide accelerometric data in all three axes: Axis 1 corresponds to the acceleration in the vertical axis (VT) (transverse/horizontal plane); Axis 2, to the medio-lateral (ML) (coronal/frontal plane); and Axis 3, to the antero-posterior (AP) (antero-posterior plane). In addition, their root mean square (RMS) was used. All accelerometers used in the study were calibrated static before use. The accelerometer measurements were configured for a time frame of 1 s. The sampling frequency selected was 100 Hz. Then, the signal was processed with a 30 Hz threshold filter before being analysed. This threshold is effective to eliminate the noise of the signal. The noise can come from the recording system itself if it is not properly fixed to the user (an aspect that must be solved with the previous calibration of the device and its proper fixing). Another origin of noise may be the selected sampling frequency, which should not exceed 50 Hz for the study of human movement, nor be too low, which may skew data collection. To eliminate this possible source of error, post-processing and averaging the signal using cut-off filters or different statistical methods may be considered [25,26].

According to the tool designed by Leirós-Rodríguez et al. [24], the study variables of static equilibrium in adult women are: 

(a) During monopodal balance on mat with open eyes: the maximum and mean values of the medio-lateral axis and RMS.

(b) During monopodal balance with open eyes: the maximum and mean values of the antero-posterior axis and RMS.

(c) During monopodal balance with closed eyes: maximum value of the antero-posterior axis and maximum and mean values of RMS.

The tests were performed while subjects were wearing socks (no shoes) and comfortable clothing, allowing them to perform the tests comfortably. The accelerometer was placed directly on the skin at the height of the spinous process of the fourth lumbar vertebra. The device was secured with an adjustable belt and hypoallergenic adhesive tape to ensure that it did not move independently to the subject’s trunk during the test. Participants performed a battery of static equilibrium tests three times. The randomised trials were: monopodal balance with eyes closed, monopodal balance with eyes open and monopodal balance on mat with eyes open [24]. 

The tests were repeated three times, separated by intervals of 30 s, to prevent the effect of lower limb muscle fatigue [27]. The mean of the duration and accelerations of three replicates was used for the analysis. During the open-eyes test, evaluators indicated participants should attempt to keep their gaze at the front but they were not told a specific point at which stare. The unstable surface test was performed on a cushioned surface (mat) with a density of 30 kg/m^3^ and dimensions of 150 cm × 100 cm × 10 cm. Participants were told that, if they suffered an imbalance while in a monopodal stance that required them to use their other leg for support, they should attempt to recover the requested position in the shortest time possible. All participants were instructed to choose the leg on which to make the support. For that, they were allowed to make previous attempts to make the selection (which they had to respect for all the tests). 

In accordance with the Declaration of Helsinki (rev. 2013), all women signed informed consent prior to their participation in the study. This research obtained ethical approval from the Commission of Ethics of the Faculty of Sciences of Education and Sport of the University of Vigo (Spain) (code: 3-0406-14).

### 2.3. Statistical Analysis

For the analysis of the results, the sample was divided into six age groups: G1 between 51 and 55 years (n = 87), G2 between 56 and 60 years (n = 72), G3 between 61 and 65 years (n = 85), G4 between 66 and 70 (n = 92), G5 between 71 and 75 years (n = 87), and G6 between 76 and 80 years (n = 73).

For the construction of the box charts and the calculation of the accelerometric reference values, the chronological age of the participants was established as the explanatory variable (years) and the accelerations recorded as the response variable (gravitational unit or g). 

To obtain more accurate accelerometric data, a wide range of percentiles was established for the response variable, taking the proposal included in the study for the development of growth standards in children of the WHO Multicentre Grow Reference Study Group as a model [28]. Extreme outliers were removed from the sample according to the criterion x < Q (25) − 3 × IQR and x < Q (75) + 3 × IQR (where IQR is the interquartile range) so as not to excessively affect the most extreme percentiles of the distributions.

For the evaluation of normality and homoscedasticity, the hypothesis tests of Kolmogórov–Smirnov and Levene were used, respectively. The Kruskal–Wallis test was used to confirm the observed differences in results between age groups. To verify whether the differences between the groups were significant, the analysis of variance (ANOVA) was used with the Bonferroni correction. This analysis was performed using the IBM SPSS Statistics for Macintosh software, Version 20.0 (SPSS, an IBM Company, Armon, NY).

For the construction of the percentile box charts and the calculation of the reference values in each group, Generalised Additive Models of Position, Scale and Form (GAMLSS) were applied [29]. The data distributions of the response variable (acceleration) were modelled by exponential Box–Cox power distributions (BCPE) applying a cubic splines technique as a smoothing method and the worm plots [30] for the evaluation of the goodness of the adjustment. To carry out this analysis, the “gamlss” package of the statistical software R (R Core Team, 2014) was used.

## 3. Results

As can be seen, as the age of the individuals increases, their weight and height are reduced, and consequently there is an increase in their BMI (Table 1).

The descriptive characteristics corresponding to the variable equilibrium (accelerations) in the three static equilibrium tests are shown in Table 2, Table 3 and Table 4. In all of the variables, an increase in the magnitude of the accelerations was detected as the age of the group advanced. In the six study groups, the null hypothesis of normal distribution (*p* < 0.01) and homoscedasticity (*p* < 0.01) was rejected for the response variable (accelerations). Likewise, the kurtosis values of the distributions determined values of positive asymmetry (>0.5) and leptocurtosis (>0.5) in all age groups and accelerometric variables.

The data show statistically significant differences when comparing the balance between women of different age groups (*p* < 0.01). 

The BOX charts of the percentiles of the tests (accelerations produced during the equilibrium tests) for women throughout aging are presented in Figure 2, Figure 3 and Figure 4. Here, the amplitude of the IQR can be observed, which increased as the age of the participants advanced. In addition, the IQR was greater in the variables that refer to the maximum values of the accelerations (in comparison to the variables that refer to the average values).

The obtained box graphs show similar trends in all groups. The layout of the boxes increases in magnitude and amplitude as the studied age group progresses.

## 4. Discussion

The objective of this study was to provide a normative data for postural stability in elderly women using percentile box charts and determine the values of reference for the accelerometric assessment of the static equilibrium in women according to the tool designed by Leirós-Rodríguez et al. [24]. In this study, the percentiles were defined and reference values were calculated in elderly women between 51 and 80 years of age. We found it relevant to present them as subgroups (G1–G6) for later comparisons in other investigations and for clinical professionals who want to make use of these reference values. The results obtained show the first data on normative values for this evaluation procedure, widely used in a research environment but whose implementation in clinical practice has not yet been generalised. The size of the sample used and the follow-up of an evaluation procedure widely justified in previous investigations determine the good representativeness of the results obtained. In this study, the differences between the equilibrium values measured by accelerometery for the different subgroups are highlighted; however, in the case of the comparison of some of the subgroups, the values should be taken with caution due to their size.

To date, no reference values have been published for the accelerometric assessment of equilibrium; this hinders the development and implementation of the methodology reported in this study and puts pressure on health professionals to continue using valuation scales (Berg Balance Score, Tinetti Test, ABC scale, etc.) that include a multitude of tests without providing a high sensitivity to the premature deterioration of balance. 

The relevance of the data presented in this study lies in the fact that it is possible to establish cut-off points or reference values for balance in older women from now on. These cut-off points are essential for designing clinical or epidemiological studies and even for their application in daily clinical practice in our environment. This work fills a gap which has already been described by other authors [31,32], within the integral geriatric assessment, which is the need to have normative values for the functional tests that are used in clinical practice and research.

The use of accelerometers allows characteristics about the degree of functionality of the patient or the risk that they may have of suffering a fall to be identified. In addition, it is a more objective alternative than the use of clinical assessment scales [27,33,34,35], as well as more sensitive, since it identifies alterations when they are not yet detectable through visual analysis [36,37]. 

The sensitivity of these devices to small changes in the functioning of postural control systems makes them very useful for the evaluation of results after physical exercise programs, physiotherapy and rehabilitation treatments, or for the early diagnosis of the deterioration of somatosensory degeneration [11,38,39]. 

It should be noted that the reference values obtained are subject to the eligibility conditions of the participants of this work: healthy women, with an active lifestyle and without trauma or orthopaedic conditioners. Set conditions indicate that the results obtained show the expected results in the study of the balance of a healthy adult or older woman.

The main limitations of the study are its cross-sectional design, the absence of males in the sample and the lack of results related to the middle-aged population. The most important interaction variable is the specific physical activity carried out by each of the participants and the existing variability in their lifestyles, with heterogeneous security as a consequence of the open nature of the inclusion criteria. Finally, the results of this work cannot be applied in assessments which do not follow the work protocol used in the research of Leirós-Rodríguez et al. [24], which was contrasted and validated for the assessment of accelerometric balance in adult and older women.

## 5. Conclusions

In this paper, the first percentiles and reference data for the accelerometric assessment of static equilibrium in adult and older women are presented. With this type of data, the deterioration of equilibrium and the risk of falling is assessed more accurately and effectively than with clinical balance tests. The values obtained can be used to assess changes in static equilibrium due to aging, trauma and orthopaedic alterations that may alter postural stability or neurodegenerative processes which increase the risk of falling. In addition, the possibility of projecting the results can contribute to improving the quality of medical treatments and physiotherapy for improving balance.

## Figures and Tables

**Figure 1 sensors-20-00940-f001:**
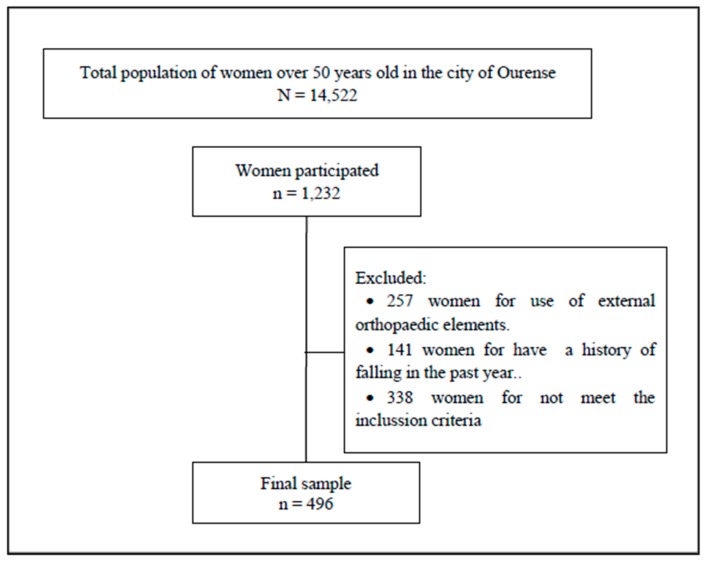
Consort flow diagram.

**Figure 2 sensors-20-00940-f002:**
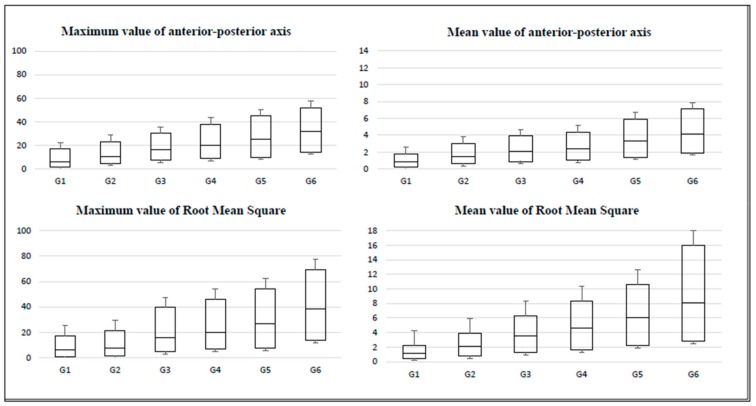
Box charts for the monopodal balance test with open eyes by age group.

**Figure 3 sensors-20-00940-f003:**
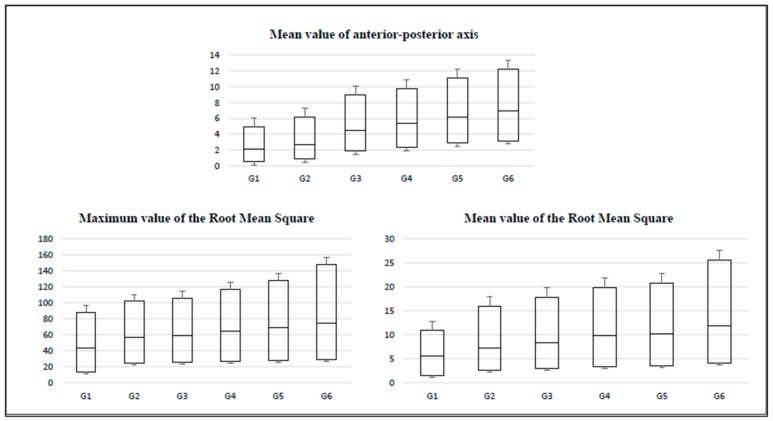
Box charts for the monopodal balance test with closed eyes by age group.

**Figure 4 sensors-20-00940-f004:**
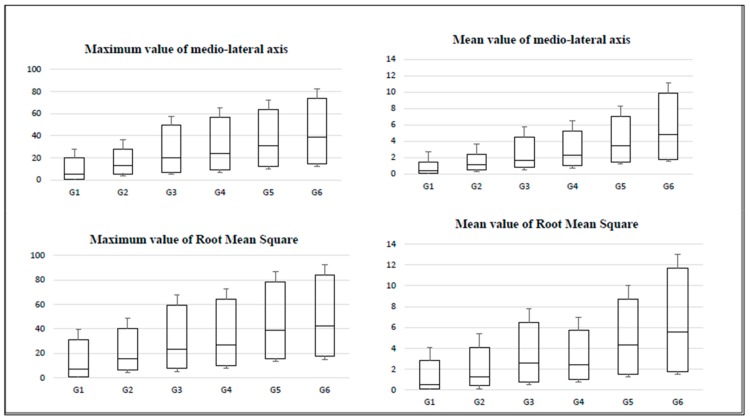
Box charts for the monopodal balance test on unstable surface (mat).

**Table 1 sensors-20-00940-t001:** Descriptive statistics of anthropometric variables.

Age Group	N	Age (years)	Weight (kg)	Height (cm)	Body Mass Index (kg/m^2^)
All	496	68.8 ± 10.4	65.6 ±10.1	153.9 ± 5.4	27.6 ± 4.1
G1 (51–55 years)	87	53.4 ± 4.4	63 ± 7.6	155.6 ± 5	26 ± 3.3
G2 (56–60 years)	72	57.4 ± 4.3	64 ± 6.5	154.6 ± 6	26.8 ± 5.3
G3 (61–65 years)	85	64.2 ± 2.7	66.4 ± 11.1	154 ± 5.5	28 ± 4.7
G4 (66–70 years)	92	68.4 ± 3.8	63.8 ± 9.6	152.9 ± 6.1	29 ± 6.3
G5 (71–75 years)	87	74.2 ± 4.6	66.5 ± 10.1	151.8 ± 5.2	28.3 ± 3.2
G6 (76–80 years)	73	77.6 ± 2.2	68.1 ± 11.7	151.3 ± 4.2	29.2 ± 1.8

**Table 2 sensors-20-00940-t002:** Percentiles and descriptive statistics for the monopodal balance test with open eyes by age group.

Variable	G1 (n = 87)	G2 (n = 72)	G3 (n = 85)	G4 (n = 92)	G5 (n = 87)	G6 (n = 73)
**Maximum value of anterior-posterior axis**						
Mean ± standard deviation	4.4 ± 6	6.2 ± 8.7	8.5 ± 8.2	9.1 ± 9.1	13.7 ± 9.9	13.9 ± 9.5
Kurtosis	4.9	4.4	4.4	5.4	2.4	2.2
Percentile 25	2	5	7.3	8.8	10.2	14.6
Percentile 50 (median)	4.2	5.7	9.5	11.6	15.2	17.5
Percentile 75	11	12.7	13.5	17.6	19.8	20
Interquartile range	9	7.7	6.2	8.8	9.6	5.4
**Mean value of anterior-posterior axis**						
Mean ± standard deviation	0.4 ± 0.8	0.5 ± 0.8	0.8 ± 1.1	1 ± 1.3	1.5 ± 1.2	1.8 ± 1.5
Kurtosis	13.4	5.7	14.9	5.2	3.6	2.2
Percentile 25	0.2	0.6	0.9	1	1.4	1.9
Percentile 50 (median)	0.7	0.9	1.2	1.4	1.9	2.2
Percentile 75	0.9	1.5	1.8	2	2.6	3
Interquartile range	0.7	0.9	0.9	1	1.2	1.1
**Maximum value of the Root Mean Square of accelerations**						
Mean ± standard deviation	7.8 ± 9.5	9.6 ± 12.3	15.3 ± 12.9	16.8 ± 14.2	23 ± 15	25 ± 17.2
Kurtosis	4.1	5.4	2.5	3.4	3.6	2.3
Percentile 25	0.8	1.3	4.8	7.1	7.6	13.6
Percentile 50 (median)	5.3	6.2	11.1	12.7	19.5	25.1
Percentile 75	11.1	13.9	23.6	26.2	27.4	30.5
Interquartile range	10.3	12.6	18.8	19.1	19.8	16.9
**Mean value of the Root Mean Square of accelerations**						
Mean ± standard deviation	0.9 ± 1.4	1 ± 1.7	1.8 ± 2.4	2.2 ± 2.7	3.1 ± 2.9	4.4 ± 3.2
Kurtosis	7.4	8.2	9.5	4.8	9.6	1.7
Percentile 25	0.5	0.8	1.3	1.6	2.2	2.8
Percentile 50 (median)	0.7	1.3	2.2	3	3.9	5.3
Percentile 75	1	1.8	2.8	3.7	4.5	7.9
Interquartile range	0.5	1	1.5	2.1	2.3	5.1

**Table 3 sensors-20-00940-t003:** Percentiles and descriptive statistics for the monopodal balance test with closed eyes by age group.

Variable	G1 (n = 87)	G2 (n = 72)	G3 (n = 85)	G4 (n = 92)	G5 (n = 87)	G6 (n = 73)
**Mean value of anterior-posterior axis**						
Mean ± standard deviation	2.2 ± 2.5	2.6 ± 1.8	2.8 ± 1.4	2.9 ± 3.3	3.2 ± 2	4.4 ± 4
Kurtosis	10	5.1	3	2.9	4.5	2.5
Percentile 25	0.6	0.9	1.9	2.4	2.9	3.2
Percentile 50 (median)	1.5	1.8	2.6	3	3.3	3.8
Percentile 75	2.8	3.5	4.5	4.4	4.9	5.2
Interquartile range	2.2	2.6	2.6	2	2	2
**Maximum value of the Root Mean Square of accelerations**						
Mean ± standard deviation	30.7 ± 18.9	36.7 ± 14.3	38.2 ± 23.2	38.7 ± 21.3	41.7 ± 17.8	52.6 ± 30
Kurtosis	1.8	2.9	2.7	5.2	2.4	2.1
Percentile 25	14	24.8	25.7	26.7	27.7	29.5
Percentile 50 (median)	30.1	32.5	33.9	37.6	41	45.3
Percentile 75	43.8	44.6	46.1	52.5	59	72.7
Interquartile range	29.8	19.8	20.4	25.8	31.3	43.2
**Mean value of the Root Mean Square of accelerations**						
Mean ± standard deviation	4.5 ± 3.8	6.3 ± 4	6.4 ± 6.3	7.2 ± 4.6	7.4 ± 3.7	10.6 ± 12.1
Kurtosis	6.1	3.6	3	3	10.2	2
Percentile 25	1.5	2.6	3.1	3.4	3.6	4.1
Percentile 50 (median)	4.1	4.7	5.3	6.5	6.7	7.8
Percentile 75	5.3	8.7	9.4	10	10.4	13.6
Interquartile range	3.8	6.1	6.3	6.6	6.8	9.5

**Table 4 sensors-20-00940-t004:** Percentiles and descriptive statistics for the monopodal balance test on unstable surface (mat).

Variable	G1 (n = 87)	G2 (n = 72)	G3 (n = 85)	G4 (n = 92)	G5 (n = 87)	G6 (n = 73)
**Maximum value of medio-lateral axis**						
Mean ± standard deviation	9.5 ± 11.6	12.6 ± 13.6	18 ± 16.4	18.2 ± 16.5	22.5 ± 15.7	22.6 ± 14.1
Kurtosis	3	3.3	5.3	2.5	6.2	1.8
Percentile 25	0.4	5.7	7	9	12.2	14.3
Percentile 50 (median)	5	7	13	15	19	24.7
Percentile 75	14.3	15.3	29.3	33	32.7	34.7
Interquartile range	13.9	9.6	22.3	24	20.5	20.4
**Mean value of medio-lateral axis**						
Mean ± standard deviation	0.9 ± 1.3	1.2 ± 1.4	2 ± 2.4	2 ± 2.7	3 ± 3.8	2.9 ± 2.1
Kurtosis	4.1	3.5	5.9	6.3	12.5	1.6
Percentile 25	0.1	0.5	0.8	1	1.5	1.8
Percentile 50 (median)	0.3	0.6	0.9	1.3	1.9	3
Percentile 75	1	1.3	2.8	2.9	3.6	5.1
Interquartile range	0.9	0.8	2	1.9	2.1	3.3
**Maximum value of the Root Mean Square of accelerations**						
Mean ± standard deviation	14.4 ± 15.7	16.9 ± 16.3	22.5 ± 19.7	22 ± 18.2	26.3 ± 16.3	29.2 ± 23.2
Kurtosis	1.9	2.6	4.7	2.5	9.6	2.1
Percentile 25	0.4	6.4	7.7	10	15.7	17.4
Percentile 50 (median)	6.4	9.3	15.8	17	23.1	25.2
Percentile 75	23.8	24.8	35.8	37.4	39.5	41.6
Interquartile range	23.4	18.4	28.1	27.4	23.8	24.2
**Mean value of the Root Mean Square of accelerations**						
Mean ± standard deviation	1.4 ± 1.7	1.7 ± 1.8	2.8 ± 3	2.8 ± 3.5	3.8 ± 2.5	4.4 ± 6
Kurtosis	2.5	3.2	5	6.1	15.5	1.8
Percentile 25	0.02	0.4	0.8	1	1.5	1.8
Percentile 50 (median)	0.4	0.9	1.8	1.4	2.8	3.8
Percentile 75	2.3	2.8	3.9	3.3	4.4	6.1
Interquartile range	2.28	2.4	3.1	2.3	2.9	4.3

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
