# Peer review of "Percentiles and Reference Values for the Accelerometric Assessment of Static Balance in Women Aged 50–80 Years"

_sensors, 2020, doi:10.3390/s20030940_

Round 1
Reviewer 1 Report
First of all, I am gratefully for the opportunity to review the manuscript entitled “Percentiles and reference values for the accelerometric assessment of static balance in women aged 50-80 years”. The topic is relevant but there are some issues that should be addressed before its publication.
My first, and most important, concern is regarding the sample size and sample characteristics to address your objective. As you pointed out “the objective of this study was to construct the percentile box charts and determine 81 the values of reference for the accelerometric assessment of the static equilibrium in women”. Thus, I have the following questions:
Is the sample size enough to consider your data as reference values? A sample size calculation has been performed to be sure that the values are representative of the population? Why did you establish 1-2 days/week of physical activity and 30-90 min walking 4 days/week? This characteristics seems to be enough to consider this sample as physically active but you aimed to establish reference values to all the population of women older than 50 years old.
Other concerns:
Line 94: why 60 seconds if the duration of your tests were of 30 seconds. Line 109: which kind of filter? Lines 110-115 are too vague. Please be specific about the procedure followed to analyze your data. Line 108: it seems that you used the accelerometer data accumulated in epochs of 1-second. Why did you not use raw data to compute variables? Moreover, what variables were computed using accelerometer signals? This is not reported in methods. Line 126: in my experience doing this kind of test there are a lot of people unable to perform monopedal with eyes closed test during 30 seconds. You pointed out that subjects can used the non-support leg to recover balance if necessary. How this fact could influence the validity of the data? There are previous studies that used this procedure? There are some studies that correlate balance scores using this procedure with most common ones (subject cannot use non-support leg and if they used it they repeat the trial)? Table 2, 3, 4 change “main” for “mean”. Moreover, in table 2 and 3 AP data is reported while in table 4 only ML data is provided. Line 137-139: why this groups of age? Please justify it. Line 149: Kruskall Wallis test should be used when there are more than two groups. Line 153-158: if you did those analysis, can you report statistical results besides box-plots? Line 184: between which groups? In what variables? What are about the statistics related with this comparisons?
Author Response
Dear Editor and Reviewer of the Sensors Journal:
Thank you very much for your suggestions and contributions to improve the quality of the manuscript. Following your indications, we respond, point by point, to your comments.
Firstly, we have rewritten the "Materials & Methods" section to avoid text matches with previous ones published works.
In the text, all the modified or added sentences have been written in red to facilitate the correction by the reviewers.
a) Is the sample size enough to consider your data as reference values? Why did you establish 1-2 days/week of physical activity and 30-90 min walking 4 days/week? This characteristics seems to be enough to consider this sample as physically active but you aimed to establish reference values to all the population of women older than 50 years old.
The region of Ourense had a population of 57,543 women in 2019. To reach a confidence level of 95% and a margin of error of 5%, the participation of 382 women was established. In our study we passed the necessary sample (n = 496). In our study we passed the necessary sample (n = 496). We recognize that it is not a study with sufficient power to generalize the results but, to be able to standardize accelerometry as a method of equilibrium assessment it is necessary to publish this type of results so that they can then be compared with those of other regions or nations.
At the same time, the weekly physical activity criteria required of participating women are based on international exercise practice recommendations. Applying this criterion seeks to standardize the conditions of participation of the sample (avoiding the influence of extraneous variables) and extract standard accelerometric values or "model" that can be used as REFERENCE.
b) Line 94: why 60 seconds if the duration of your tests were of 30 seconds.
To eliminate the influence of fatigue or lack of strength in lower limbs in performing the tests. If the participants are able to maintain 60 seconds the position means that the measurement of 30 seconds should not be influenced by the tiredness of the muscles. At the same time, it is a safety measure that participants are not at risk of serious falls during the evaluation.
c) Line 109: which kind of filter?
A threshold filter. This filter changes the acceleration value recorded for a given epoch to a constant if the raw value exceeds a low threshold; raw values at or below the threshold are set to zero. By removing the variation in the recordings above the threshold value, this transformation prevents erratic fluctuations in body acceleration from influencing the measurement of the amount of movement.
We have specified the type of filter in the text.
d) Line 108: it seems that you used the accelerometer data accumulated in epochs of 1-second. Why did you not use raw data to compute variables? Moreover, what variables were computed using accelerometer signals? This is not reported in methods. Lines 110-115 are too vague. Please be specific about the procedure followed to analyze your data.
In order to analyze the specific time periods of the tests, the authors could not use the raw data, but by synchronizing with the computer, the acceleration sequences corresponding to those of the tests were selected. In this way we eliminate the rest time between tests or the time between the placement of the device and the start of the first test.
The description of the data processing for the extraction of the analyzed variables has been completed.
e) Line 126: in my experience doing this kind of test there are a lot of people unable to perform monopedal with eyes closed test during 30 seconds. You pointed out that subjects can used the non-support leg to recover balance if necessary. How this fact could influence the validity of the data? There are previous studies that used this procedure?
In our study, all participants were able to maintain the monopodal posture. This may be due to the demanding criteria of inclusion and exclusion that we define. These, at the same time, agree with the employees in previous investigations:
Leirós-Rodríguez R, Romo-Pérez V, García-Soidán JL. Validity and reliability of a tool for accelerometric assessment of static balance in women. European Journal of Physiotherapy. 2017;19(4):243-248.
Leirós-Rodríguez R, García-Soidán JL, Romo-Pérez V. Analyzing the use of accelerometers as a method of early diagnosis of alterations in balance in elderly people: A systematic review. Sensors. 2019;19(18):3883-3907.
At the same time, reducing the analysis until the participant rests the other foot on the ground for the first time can be an error because it biases the multidimensionality of the balance, not taking into account factors such as the ability to adapt to monopodal support (which is not the normal in normal conditions for the individual) or the development of neuromuscular strategies according to learning.
f) Table 2, 3, 4 change “main” for “mean”. Moreover, in table 2 and 3 AP data is reported while in table 4 only ML data is provided.
The "mean" errata has been fixed.
The justification of the inclusion of different variables for each assessment test has been added in Materials & Methods section.
g) Line 137-139: why this groups of age? Please justify it.
Taking into account the age range of the participants (51 - 80 years), the division by age groups of 5 years is sufficient to objectify differences between the groups generated, it is practical for the extraction of reference values and, at the same time , sensitive to the natural aging process.
h) Line 149: Kruskall-Wallis test should be used when there are more than two groups.
It was a mistake of writing the manuscript. The Kruskal-Wallis test was done (the Mann – Whitney U test was NOT applied).
i) Line 184: between which groups? In what variables? What are about the statistics related with this comparisons?
That sentence from the Results section has been rewritten and the explanation of the statistical analysis has been completed in Materials & Methods section.
Once again, thank you very much for the time spent and the interest shown in this work; as well as in the positive evaluations you have given of it.
Receive a warm greeting,
The authors.

Reviewer 2 Report
The topic is interesting but I think a better rationale is needed to validate the use of accelerometers.
It is necessary to clarify some questions:
Lines 50-59 highlights some limitations of using force platforms for balance assessment, and does the accelerometer placed on the lower back evaluate the movement of the CG?
During the open-eyes test, was there any indication given to fix your gaze on a target or something similar?
What is the rationale for using RMS for global acceleration instead of using RMS for the mediolateral and anteroposterior components?
In the line 109 refers to the use of a filter (30HZ), what are its characteristics?
Why not use vertical acceleration as it has also been evaluated?
Author Response
Dear Editor and Reviewer of the Sensors Journal:
Thank you very much for your suggestions and contributions to improve the quality of the manuscript. Following your indications, we respond, point by point, to your comments.
Firstly, we have rewritten the "Materials & Methods" section to avoid text matches with previous ones published works.
In the text, all the modified or added sentences have been written in red to facilitate the correction by the reviewers.
a) Lines 50-59 highlights some limitations of using force platforms for balance assessment, and does the accelerometer placed on the lower back evaluate the movement of the CG?
Yes, in line with a systematic review published in Sensors: “The placement of the devices has varied between the last vertebrae of the lumbar region and the pelvis. Although the accelerometric data have been collected regularly in the lumbar area to analyze balance, we can find diferences in the records obtained due to the di_erent mechanical needs of the area and the internal relations that it maintains with the pelvis. When the accelerometer is placed in the fifth lumbar vertebra, it collects the pressure forces at the base of the sacrum and may be influenced by the rotational movements of the sacrum.
Therefore, its placement in the fourth lumbar vertebra is recommended so as not to collect data on the mobility of the pelvic girdle”1.
Leirós-Rodríguez R, García-Soidán JL, Romo-Pérez V. Analyzing the use of accelerometers as a method of early diagnosis of alterations in balance in elderly people: A systematic review. Sensors. 2019;19(18):3883-3907.
b) During the open-eyes test, was there any indication given to fix your gaze on a target or something similar?
Yes, the indication given was "trying to keep your eyes straight ahead" but no specific point or anything similar was indicated. This aspect has been included in Materials & Methods section.
c) What is the rationale for using RMS for global acceleration instead of using RMS for the mediolateral and anteroposterior components?
Although the RMS for the mediolateral and anteroposterior axes could also have been calculated, we chose the global RMS. We based this decision on the literature published to date (including the review we cited previously), the overall RMS is the most widely used parameter and has proven to be more clinically relevant.
d) In the line 109 refers to the use of a filter (30HZ), what are its characteristics?
A threshold filter. This filter changes the acceleration value recorded for a given epoch to a constant if the raw value exceeds a low threshold; raw values at or below the threshold are set to zero. By removing the variation in the recordings above the threshold value, this transformation prevents erratic fluctuations in body acceleration from influencing the measurement of the amount of movement.
We have specified the type of filter in the text.
e) Why not use vertical acceleration as it has also been evaluated?
Because we only analyzed the accelerations in the axes that a previous investigation included in a clinical tool for the accelerometric assessment of the static equilibrium in women designed by Leirós-Rodríguez et al2.
The inclusion of different variables for each assessment test has been explained in Materials & Methods section.
Leirós-Rodríguez R, Romo-Pérez V, García-Soidán JL. Validity and reliability of a tool for accelerometric assessment of static balance in women. European Journal of Physiotherapy. 2017;19(4):243-248.
Once again, thank you very much for the time spent and the interest shown in this work; as well as in the positive evaluations you have given of it.
Receive a warm greeting,
The authors.

Reviewer 3 Report
Sensors-680239
Percentiles and reference values for the accelerometric assessment of static balance in women aged 50-80 years
General Comments:
The manuscript attempts to provide normative balance data in elderly women by using cheaper and more feasible accelerometer devices. The study adds value to the current literature in postural stability and fall prevention in elderly population. There are some edits and suggestions that need to be made by the authors before the manuscript is suitable for publication. I have mentioned them under specific comments and minor comments.
Specific Comments:
Abstract:
Line 21 is in completely written. Needs to be revised.
Line 22: Change to three tests (plural).
Line 23: the use of the words monopodal vs. monopedal is purely based on the country of origin of the paper. The authors may want to verify this, and as long as readers are informed, it avoid confusion to readers all across the world.
Line 22-23 grammatically incorrect and needs to be revised.
Introduction:
The introduction provides a good summary of postural control and the need for less expensive measurement tools for postural stability and balance. However the addition of the following postural stability tools and measures using Neurocom Equitest, Force Plates and novel cheaper balance platforms such as BTrackS can help aid the introduction. These are especially done in elderly populations.
Turner, A., Chander, H., & Knight, A. (2018). Falls in geriatric populations and hydrotherapy as an intervention: a brief review. Geriatrics, 3(4), 71. Goble, D. J., & Baweja, H. S. (2018). Normative data for the BTrackS balance test of postural sway: results from 16,357 community-dwelling individuals who were 5 to 100 years old. Physical therapy, 98(9), 779-785. Dabbs, N. C., MacDonald, C. J., Chander, H., Lamont, H. S., & Garner, J. C. (2014). THE EFFECT OF WHOLE-BODY VIBRATION ON BALANCE IN ELDERLY WOMEN. Medicina Sportiva, 18(1).
Lines 81-82: Suggest revising these statements. I don’t think the purpose is construct percentile box charts. The purpose is to provide a normative data for postural stability in elderly women using percentile box charts.
Methods:
Please move lines 117 – 120 about informed consent a little ahead in the methods sections.
Line 127: The reference for the balance tests performed helps, but need more justification as to why only these three specific tests were used.
Discussion:
Please restate the purpose again as the starting of the discussion section.
Lines 211-212: Other forms of normative data for balance are available. Just need to be acknowledged.
The discussion needs to be elaborated a bit. Please focus on the advantages and disadvantages of using accelerometer to measure balance. Such as, it is cheaper and faster, but may not be as precise as force platforms or other clinical balance machines such as Neurocom, biodex etc. that provide manipulations of sensory systems in measuring balance. The limitations listed on the manuscript are fines, but readers need to be aware of the advantages and disadvantages of using the actigraph accelerometer for postural stability measurements.
One of the biggest advantage is that accelerometers can be worn by elderly population and can be used for continuous monitoring to emphasize fall prevention in elderly.
Combine the 2 paragraphs into one in the conclusion section.
Minor Comments:
The reference style is not correct for MDPI papers and needs to be changed appropriately.
Author Response
Dear Editor and Reviewer of the Sensors Journal:
Thank you very much for your suggestions and contributions to improve the quality of the manuscript. Following your indications, we respond, point by point, to your comments.
Firstly, we have rewritten the "Materials & Methods" section to avoid text matches with previous ones published works.
In the text, all the modified or added sentences have been written in red to facilitate the correction by the reviewers.
a) ABSTRACT: Line 21 is in completely written. Needs to be revised. Line 22: Change to three tests (plural). Line 23: the use of the words monopodal vs. monopedal is purely based on the country of origin of the paper. The authors may want to verify this, and as long as readers are informed, it avoid confusion to readers all across the world. Line 22-23 grammatically incorrect and needs to be revised.
All these grammatical errors have been corrected.
b) INTRODUCTION: The introduction provides a good summary of postural control and the need for less expensive measurement tools for postural stability and balance. However the addition of the following postural stability tools and measures using Neurocom Equitest, Force Plates and novel cheaper balance platforms such as BTrackS can help aid the introduction. These are especially done in elderly populations.
Turner, A., Chander, H., & Knight, A. (2018). Falls in geriatric populations and hydrotherapy as an intervention: a brief review. Geriatrics, 3(4), 71.
Goble, D. J., & Baweja, H. S. (2018). Normative data for the BTrackS balance test of postural sway: results from 16,357 community-dwelling individuals who were 5 to 100 years old. Physical therapy, 98(9), 779-785.
Dabbs, N. C., MacDonald, C. J., Chander, H., Lamont, H. S., & Garner, J. C. (2014). The effect of whole-body vibration on balance in elderly women. Medicina Sportiva, 18(1).
We have completed the Introduction section explaining the devices that you recommend and we have used the studies you indicated as bibliographical references.
c) Lines 81-82: Suggest revising these statements. I don’t think the purpose is construct percentile box charts. The purpose is to provide a normative data for postural stability in elderly women using percentile box charts.
We have rewritten the objectives of the study.
d) METHODS: Please move lines 117 – 120 about informed consent a little ahead in the methods sections. Line 127: The reference for the balance tests performed helps, but need more justification as to why only these three specific tests were used.
We have rewritten the Materials & Methods section, we have added the modifications you indicate and we have better explained the inclusion of each of the tests and the study variables.
e) DISCUSSION: Please restate the purpose again as the starting of the discussion section.
We have included a sentence with the objectives of the study at the beginning of the Discussion section.
f) Combine the 2 paragraphs into one in the conclusion section.
That has been corrected.
g) The reference style is not correct for MDPI papers and needs to be changed appropriately.
We have corrected the reference style according to MDPI papers.
Once again, thank you very much for the time spent and the interest shown in this work; as well as in the positive evaluations you have given of it.
Receive a warm greeting,
The authors.

Round 2
Reviewer 1 Report
In my opinion there are some metodological flaws that prevent to address the objectives of the manuscript. Moreover, in my opinion the changes and justifications of authors are not enought to consider the manuscript suitable to be published in Sensors.